# Preparation of Hybrid Films Based in Aluminum 8-Hydroxyquinoline as Organic Semiconductor for Photoconductor Applications

**DOI:** 10.3390/s23187708

**Published:** 2023-09-06

**Authors:** María Elena Sánchez Vergara, Luis Alberto Cantera Cantera, Citlalli Rios, Roberto Salcedo, Octavio Lozada Flores, Ateet Dutt

**Affiliations:** 1Facultad de Ingeniería, Universidad Anáhuac México, Avenida Universidad Anáhuac 46, Col. Lomas Anáhuac, Huixquilucan 52786, Estado de México, Mexico; 2Instituto de Investigaciones en Materiales, Universidad Nacional Autónoma de México, Circuito Exterior s/n. C.U., Mexico City 04510, Mexico; 3Facultad de Ingeniería, Universidad Panamericana, Augusto Rodin 498, Insurgentes Mixcoac, Mexico City 03920, Mexico

**Keywords:** hydroxyquinoline, DFT calculations, hybrid film, optical properties, organic photoconductor

## Abstract

In the present work, we have investigated an organic semiconductor based on tris(8-hydroxyquinoline) aluminum (AlQ_3_) doped with tetracyanoquinodimethane (TCNQ), which can be used as an organic photoconductor. DFT calculations were carried out to optimize the structure of semiconductor species and to obtain related constants in order to compare experimental and theoretical results. Subsequently, AlQ_3_-TCNQ films with polypyrrole (Ppy) matrix were fabricated, and they were morphologically and mechanically characterized by Scanning Electron Microscopy, X-ray diffraction and Atomic Force Microscopy techniques. The maximum stress for the film is 8.66 MPa, and the Knoop hardness is 0.0311. The optical behavior of the film was also analyzed, and the optical properties were found to exhibit two indirect transitions at 2.58 and 3.06 eV. Additionally, photoluminescence measurements were carried out and the film showed an intense visible emission in the visible region. Finally, a photoconductor was fabricated and electrically characterized. Applying a cubic spline approximation to fit cubic polynomials to the J-V curves, the ohmic to SCLC transition voltage $V_{\mathrm{ON}}$ and the trap-filled-limit voltage $V_{\mathrm{TFL}}$ for the device were obtained. Then, the free carrier density and trap density for the device were approximated to $n_{0}=4.4586\times 10^{19}\frac{1}{m^{3}}$ and $N_{t}=3.1333\times 10^{31}\frac{1}{m^{3}}$, respectively.

## 1. Introduction

The development of optoelectronic and photovoltaic devices based on organic semiconductors has drawn attention due to their promising applications [1,2,3]. Specifically, an organic optocoupler (OOC) consists of three main components: an organic photoemitter, an insulator and an organic photodetector [4]. Such an organic photodetector, also known as a photosensor, can be divided into organic photoresistors, organic photodiodes or organic phototransistors [4,5]. These organic semiconductor devices possess a photoconductive effect, which modifies their conductivity due to the incidence of light; i.e., when a voltage is applied, a current flows and the incident light on the device will cause an increase in this current [6,7]. The current flowing in a semiconductor in the absence of some kind of light is called dark current, whereas the flowing current produced by an incident light is called photon-induced current or photocurrent [7]. When an organic semiconductor exhibits the photoconductive effect, it may be classified as an organic photoconductor (OPC) [4,7].

Organic semiconductors require a conjugated structure to participate effectively in electric conductivity [1,2]. The surface of their π orbitals should stabilize the electronic charge by resonant effect and facilitate intermolecular interactions by overlapping orbitals. For this reason, the correct selection of the organic base structure is a determining aspect in the design of new organic semiconductors. Small π-conjugated molecules are excellent candidates for assembling different types of organic semiconductors due to their attractive features, such as well-defined chemical structure, easily modified structural layouts, fine-tuning of their energy levels and excellent reproducibility [8,9,10,11,12]. Small π-conjugated molecules can be processed in solution [11]; besides, due to their low molecular weight [8,13], they can also be deposited by vacuum thermal evaporation for assembling ordered layers that improve the performance of the final device. Vacuum thermal deposition allows to produce devices with controlled multilayers and stability ideal for commercialization. However, small π-conjugated molecules with sufficiently high thermal stability and better light harvesting capability are required [13]. For this reason, solution deposition methods are also used in processing small molecules either in solution or dispersion [14,15,16,17]. This latter feature provides several advantages, such as the possibility of preparing thin films with insoluble molecules showing high chemical stability, which, when forming part of photovoltaic and optoelectronic devices, will enhance their performance without degrading easily.

Metal quinolines (MQs) are small molecules that have been used in organic optoelectronic devices [18,19,20,21,22] due to their ability to form thin films in solution [18,23,24,25] and by evaporation [19,22,26,27] since they exhibit high chemical and thermal stability. MQs also offer easy color tunability with simple synthesis [28,29,30,31], magnetism properties [32] and electron transport behavior [23,28,33]. Moreover, the fluorescence of MQs has led to their classification as an important class of electro-luminescent/electron transport materials [23]. It is also known that changing the central metal ion affects the luminescence peak position of the MQs [18,20,21,22,23,34]. Above the MQs family, tris(8-hydroxyquinoline) aluminum (Alq_3_) is the most popular MQs semiconductor due to its strong luminesce, high electron mobility and, in addition, due to its low cost and simple technology of fabrication [26,35]. Alq_3_ is a small molecule with conjugated π-electron systems and has a non-centrosymmetric crystal structure [35] with ∝, β, γ, δ and ε crystalline phases [35,36,37,38,39]. Alq_3_ has become used to fabricate thin films by both the solution process and by using vacuum thermal deposition [26,39]; therefore, a considerable number of studies are focused on the nucleation and growth of Alq_3_ films [40], as well as on their surface topography [41], their optical properties and their energy band gap under indirect and direct conditions [26]. Additionally, it should be considered that many studies have investigated the applications of Alq_3_ films on organic light-emitting devices (OLEDs) [24,35,42], display devices and systems [43] and organic photovoltaic cells [44]. However, only a few studies have been devoted to understanding the current–voltage characteristics in order to manipulate doping and improve the performance of Alq_3_, the latest mainly focusing on electrical conduction mechanism analysis, such as transition from ohmic to space-charge-limited current (SCLC) regimes [45,46], as well as on the estimation of electrical parameters, such as trap factors, free carrier density, ohmic to SCLC transition voltage and trap-filled-limit voltage [47,48,49].

Although space-charge-limited current theory dates to the 1940s through the work of Mott and Gurney [50], the study of ohmic and space-charge-limited currents in semiconductors is the most widely used to explain the mechanisms of charge transport and carrier trapping. In the case of the SCLC regime, it takes place when the electrode injects more carriers than those the material can transport and is modeled by a quadratic equation. On the other hand, the ohmic regime exhibits a linear behavior that arises when the amount of injection is low compared to thermally generated carriers and impurities [45,46].

The present work aims to employ Alq_3_ as a small molecule to produce active layers of a photoconductor device showing a one-layer structure formed by Alq_3_ doped with tetracyanoquinodimethane (Alq_3_-TCNQ). TCNQ is an electronic acceptor capable of forming stable radicals, with valence electrons located above and below the median plane of the molecule in delocalized π orbitals [51,52]. TCNQ favors the formation of molecular anisotropic blocks, where conduction channels are generated. Subsequently, the Alq_3_-TCNQ-doped semiconductor was embedded in a polypyrrole (Ppy) polymeric matrix to fabricate a hybrid active layer with a dispersed heterojunction architecture. In this layer, there is high surface contact between Alq_3_-TCNQ and the Ppy matrix, which favors the dissociation of excitons. Furthermore, Ppy was chosen due to its conductive properties and moderate band gap caused by its heteroatoms and π-conjugated system [53]. Ppy possesses a unique combination of environmental stability and structural versatility that allows it to be used as a matrix in the fabrication of heterojunction hybrid films. The (Alq_3_-TCNQ):Ppy film was structurally and morphologically characterized, and its optical properties were also evaluated. With respect to the theoretical calculations, a DFT method was used to optimize the structure of Alq_3_-TCNQ semiconductor complexes; their IR spectra were calculated and compared to those experimentally obtained. Additionally, their frontier molecular orbitals were determined, and the corresponding bandgap values were calculated with these results. Finally, a simple device was fabricated with the hybrid film, and their electrical characteristics were investigated by current-voltage (I-V) measurements under different lighting conditions. The motivation to study the electrical behavior of (Alq_3_-TCNQ):Ppy hybrid films under different lighting conditions is related to determining whether their optical properties affect the amount of electric current transported through this film. The theoretical model can explain the conduction mechanisms in the device. Also, based on the current-voltage measurements of the device and its approximation by cubic splines, the ohmic to SCLC transition voltage $V_{\mathrm{ON}}$ and the trap-filled-limit voltage $V_{\mathrm{TFL}}$ were obtained.

## 2. Theoretical Calculations

The structural characterization of the parent molecules Alq_3_ and TCNQ and the Alq_3_-TCNQ semiconductor was carried out using the B3PW91 hybrid method contained in Gaussian 16 Package [54] and the 6-31G basis set. The method combines Becke’s gradient corrections [55] for exchange and Perdew–Wang’s for correlation [56]. Frequency calculations were carried out at the same level of theory in order to confirm that the optimized structures were at the minimum of the potential energy surface.

## 3. Materials and Methods

Tris-(8-hydroxyquinoline)aluminum (Alq_3_; C_27_H_18_AlN_3_O_3_), 7,7,8,8-tetracyanoquinodimethane (TCNQ; C_12_H_4_N_4_) and polypyrrole (Ppy) were obtained from Sigma Aldrich (Sigma-Aldrich, St. Louis, MO, USA) and required no further purification. Afterwards, an Alq_3_-TCNQ-doped semiconductor was obtained by the dissolution of 106 mg (0.3 mmol) of Alq_3_ and 102 mg (0.5 mmol) of TCNQ in 20 mL of ethanol. Doping was carried out for 30 min at 425 K in a heated reactor Monowave 50 (Anton Paar México, S.A. de C.V. Hidalgo, México) with a pressure sensor. The reactor is operated with a borosilicate glass vial and manually closed by a cover with an integrated pressure (0–20 bar) and temperature sensor. The system was cooled and brought to atmospheric pressure; the Alq_3_-TCNQ was filtered, washed with ethanol and dried in a vacuum. The Alq_3_-TCNQ semiconductor was subsequently deposited on Ppy to form a composite film onto different substrates: glass, high-resistivity monocrystalline n-type silicon wafers (c-Si) and coated glass slides with Fluorine-Tin-Oxide (FTO). The glass and FTO substrates were previously cut and washed consecutively in an ultrasonic bath with chloroform, ethanol and acetone solvents. The silicon substrates, however, were cut and later washed with “p” solution (10 mL HF, 15 mL HNO_3_ and 300 mL H_2_O). The film was fabricated from a simple dispersion with 5 mL of the Ppy and Alq_3_-TCNQ from a dilution of 10 wt% in *p*-cresol. The distribution Alq_3_-TCNQ with Ppy was dispersed using the G560 shaker of Scientific Industries Vortex-Genie (Bohemia, NY, USA). A syringe was used to apply 0.6 mL on the substrate surface to obtain the films and then spread the dispersion. Subsequently, the films were brought to 55 °C for 5 min in the drying oven Briteg SC-92898 (Briteg Instrumentos Científicos, S.A de C.V. Mexico City, Mexico). The last accelerated the film fabrication and prevented the samples from swelling and cracking during drying. An FTIR spectroscopy analysis was performed to verify the main functional groups of the Alq_3_-TCNQ semiconductor using a Nicolet iS5-FT spectrometer (Thermo Fisher Scientific Inc., Waltham, MA, USA). Morphological and topographical characteristics were, respectively, investigated with a ZEISS EVO LS 10 scanning electron microscope (SEM, Carl Zeiss AG. Jena, Germany) and with a Nano AFM atomic force microscope (AFM, Nanosurf AG, Liesta, Switzerland) using a Ntegra platform for the (Alq_3_-TCNQ):Ppy films deposited on both glass and silicon substrates. Additionally, the films on glass were examined by X-ray diffraction analysis (XRD) using the θ–2θ technique in a Rigaku Miniflex 600 diffractometer (Rigau Corporation, Tokyo, Japan) with Cu Kα (λ = 1.5406 Å) at 40 kV, 20 mA. Thickness measurements were carried out using the ZIGO-NEXVIEW profilometry technique. The UV-Vis spectra in glass films were obtained in the 200-1100 nm wavelength range using a UV-Vis 300 Unicam spectrophotometer (Thermo Fisher Scientific Inc., Waltham, MA, USA). Photoluminescence measurements were carried out using He-Cd 325 nm laser (Kimmon Koha Co., Ltd., Tokyo, Japan) at room temperature on an optical table. Finally, the electrical behavior was procured through current–voltage (I-V) measurements in a Keithley 4200-SCS-PK1 auto-ranging pico-ammeter (Tektronix Inc., Beaverton, OR, USA). The evaluation was performed on the FTO/(Alq_3_-TCNQ):Ppy/Ag device (Figure 1a); Ag was deposited on top of the layers to act as a cathode, and the FTO acted as anode. I-V curves were obtained under darkness, white, red, orange, yellow, green, blue and ultraviolet lights in a range of −1.2 to 1.2 V at room temperature. The principal studies of the current-voltage (I-V) characteristics in semiconductors are the ohmic and space-charge-limited currents (SCLC); therefore, several investigations, such as [45,46,57,58,59], have identified a typical current density-voltage curve, as shown in Figure 1b.

Two regimes can be identified in Figure 1b; the first one is the ohmic regime at low voltages that can be expressed by
(1)$$J_{\Omega }=qn_{0}\mu_{n}\frac{V}{d_{s}},$$
when the voltage is increased, the second regime is quadratic, called the space-charge-limited current (SCLC) regime, defined by
(2)$$J_{\mathrm{SCLC}}=\frac{9}{8}\mu_{n}\varepsilon_{0}\varepsilon_{r}\frac{V^{2}}{d_{s}^{3}},$$
where J denotes the current density, q is the electronic charge, $n_{0}$ is the free carrier density, $\mu_{n}$ is the electron mobility, V is the applied voltage, $d_{s}$ is the thickness of the sample, $\varepsilon_{0}$ is the vacuum permittivity and $\varepsilon_{r}$ is the dielectric constant. Generally, a SCLC regime also occurs at high voltages but with a higher growth rate than the quadratic regime. In addition to the ohmic and SCLC regimes, it has been identified the ohmic to SCLC transition voltage denoted by $V_{\mathrm{ON}}$, and the trap-filled-limit voltage denoted by $V_{\mathrm{TFL}}$ determines the change from one quadratic regime to another. These transition voltages are defined by
(3)$$V_{\mathrm{ON}}=\frac{8}{9}\frac{qn_{0}}{\varepsilon_{0}\varepsilon_{r}}d_{s}^{2},$$
(4)$$V_{\mathrm{TFL}}=\frac{qN_{t}}{2\varepsilon_{r}}d_{s}^{2},$$
where $N_{t}$ is the trap density. Notice that, when the transition voltages are determined from Equations (3) and (4), the free carrier density and the trap density can be calculated.

## 4. Results and Discussion

### 4.1. DFT Calculations

To determine the feasibility of doping the electronic donor Alq_3_ with the electronic acceptor TCNQ, the molecular geometry of both precursor compounds and the organic semiconductor was optimized by using the DFT method. The energy values of the molecular orbitals HOMO (highest occupied molecular orbital) and LUMO (lowest unoccupied molecular orbital) for hydroxyquinoline Alq_3_ are −5.1 and −1.95 eV, respectively, and the band gap value is 3.15 eV. These results are in the same order of magnitude as those calculated by Sevgili et al. [24] for aluminum 8-hydroxyquinoline microbelts and microdots. For the TCNQ, the calculated HOMO value is −7.45 eV, the LUMO has a value of −4.95 eV and the bandgap is 2.5 eV. It should be considered that the band gaps obtained for both precursor species are significant to be considered efficient organic semiconductors. The control of the HOMO–LUMO energy gap of organic semiconductors is fundamental for the effective transport of charges. To understand the interaction between hydroxyquinoline and TCNQ, two possibilities were studied. The first one considers the interaction between central oxygens of hydroxyquinoline and the aromatic ring of TCNQ (Hal-TCNQ1). In a second case, the interaction between the hydrogen atoms of one of the rings of hydroxyquinoline and the nitrogen atoms of TCNQ (Hal-TCNQ2) was studied. These two interactions are the unique stable possibilities of dispersion forces interactions and are hydrogen bonds. Figure 2 shows the optimized structures of the mentioned cases, and the mentioned hydrogen bonds are clearly demonstrated.

Figure 3a,b show the frontier molecular orbitals for Hal-TCNQ1; HOMO is located over hydroxyquinoline and LUMO over TCNQ; this result was expected because TCNQ is known for being an excellent electron acceptor [60]. The donor–acceptor behavior of the species is the source of the interaction between them; the HOMO and LUMO values are 5.14 and 4.87 eV, respectively, and the band gap value yields 0.271 eV. In order to measure the importance of the interaction between the molecules, the Wiberg index was calculated. Table 1 shows the distance between the interacting atoms and the Wiberg index values. The calculations showed that three hydrogen bonds were formed between hydrogen atoms of hydroxyquinoline and TCNQ. On the other hand, Figure 3c,d show the frontier molecular orbitals of Hal-TCNQ2; again, HOMO is located over hydroxyquinoline and LUMO over TCNQ. In this case, HOMO yields 5.44 eV and LUMO 4.47 eV, whereas the band gap value is 0.968 eV. Again, the Wiberg index was calculated to analyze the interaction between atoms. Three hydrogen bridges were localized; Table 1 shows the distance and the Wiberg index for the involved atoms. It is observed that, for the interaction O(51)-H(70), the Wiberg index is an order of magnitude higher and exhibits the smallest distance; thus, this is the main bond leading the interaction. Based on the information, it is possible to conclude that both molecules constitute an electronic complex by means of hydrogen bonds, its energy gap decreases and the semiconductor behavior is notably improved with the formation of this complex. An explanation for this behavior is that the intermolecular interaction (the hydrogen bond) directly arises from the HOMO of hydroxyquinoline and the LUMO of TCNQ, whose energy values are very close.

It is important to consider that the values of 0.271 and 0.968 eV for Hal-TCNQ1 and Hal-TCNQ2 correspond to low band gap semiconductors [61,62]. Apparently, the donor-acceptor approach introduced by Havinga et al. [62] is the cause of the significant decrease in the band gap value. The position of the HOMO and LUMO levels and the band gap of the resulting doped semiconductor are determined by the hybridization of the corresponding frontier orbitals of the Alq_3_ and TCNQ units (see Figure 3). Therefore, based on the DFT calculations, carrying out the experimental doping of Alq_3_ with TCNQ is convenient. Additionally, the IR spectra were theoretically obtained for both HAl-TCNQ1 and hAl-TCNQ2 (see Figure 4), and the results provide a hint of the feasibility of carrying out the experimental doping of the semiconductor Alq_3_-TCNQ.

### 4.2. Fabrication and Characterization of Hybrid Film

After the doping, the Alq_3_-TCNQ was characterized by IR spectroscopy in the KBr pellet. The results were compared to those theoretically obtained; both spectra are shown in Figure 4. Characteristic peaks of Alq_3_ bands at 400–650 cm^−1^ can be related to the stretching vibration of the metal ion with the attached ligand, while the peak at 1582 cm^−1^ is associated with the quinoline group of Alq_3_ [21,63,64,65,66]; additionally, aromatic stretching C=C (1607 cm^−1^) is observed in the spectrum. C-C (1469 cm^−1^) and C-C-H bending vibrations (1169 cm^−1^) are also present, while peaks at 781 and 646 cm^−1^ are associated with in-plane ring deformations [21,63]. On the side of the TCNQ dopant, the signal is observed at 2215 cm^−1^, referring to the CN-stretching mode of the cyano groups [67]. A more detailed analysis is presented in Table 2 to further understand the spectra. The good agreement between the experimental IR spectrum and that theoretically obtained indicates that the calculations yield an excellent approach to the experimental study. However, the hAl-TCNQ2 IR spectrum seems to be more similar to that experimentally obtained. The observed differences between experimental and theoretical spectra are mainly attributed to the conditions established for DFT calculations; indeed, only the interaction between a hydroxyquinoline molecule and a TCNQ molecule as isolated species is considered; i.e., there is no other kind of interaction either with similar molecules or with impurities; thus, this spectrum is very clean with respect to its experimental counterpart. The multiple interactions between Alq_3_ molecules and their dopant are reflected in the experimental spectrum; such interactions cause both bathochromic and hypsochromic shifts. An IR spectrum of (Alq_3_-TCNQ): Ppy film was carried out to verify that, after preparing the hybrid film, the doped semiconductor suffered no degradation. The results are shown in Table 2. As expected, no degradation was found.

SEM analysis was carried out to analyze the morphology of the (Alq_3_-TCNQ):Ppy film, and Figure 5a shows the 250× microphotography of the hybrid film. As can be seen, the film is made up of clusters of various sizes that are distributed along the entire surface. It is a heterogeneous film regarding the shape and size of the clusters formed. The XRD pattern of (Alq_3_-TCNQ):Ppy film is shown in Figure 5b and exhibits apparent broad diffused diffraction reflections at 2θ regions of 16–38°. No sharp Bragg peaks that could be related to a crystalline structure were presented. This indicates that the clusters assembled in the film and the film itself (see Figure 5a) are essentially amorphous. To complement the information, the (Alq_3_-TCNQ):Ppy film was studied by AFM, and, as observed in Figure 5c, an irregular topography integrated by crests and valleys of different sizes is presented. This topography is reflected in a root mean square (RMS) of 23.04 nm and an arithmetic mean roughness (Ra) of 15.03 nm. Moreover, considering a maximum applied force of 990 N, the mechanical properties of the hybrid film are (i) the unitary deformation (ε) = 0.744, (ii) the maximum stress (σ_max_) = 8.66 mPa and (iii) the Knoop microhardness (HK) = 0.0311. One of the most important features for applications in organic optoelectronic devices is high mechanical resistance, which, according to the obtained results, is exhibited by (Alq_3_-TCNQ):Ppy film.

### 4.3. Optical Characterization of the Device

Aside from controlled band gap, active materials for electronic and photonic applications should demonstrate appropriate absorption and/or emission properties, highest occupied and lowest unoccupied molecular orbital energy levels and charge transport properties. The UV-Vis spectra of the semiconductor film are shown in Figure 6. When the purpose is to use the (Alq_3_-TCNQ):Ppy film in optoelectronic and photovoltaic devices, then a deep understanding of its optical behavior is required. Figure 6a,b show the absorbance and transmittance, respectively, for the films. In the absorbance spectrum of Alq_3_-TCNQ film, an absorption band of 375 nm is observed. It has been found that this energy absorption is due to the electron transition from the HOMO in the Alq_3_ to the LUMO in the TCNQ [22]. In another context, an observed small band at 860 nm is also due to charge transfer with TCNQ. Regarding transmittance, the bands in the film become more noticeable and appear in the 570 to 800 regions. Another feature to highlight in the spectrum of Figure 6b is the high transmittance at a larger wavelength of λ > 900 nm; the transparency of around 80% at high wavelengths coincides with that obtained for both the Alq_3_ precursor [24] and other hydroxyquinolines, such as zinc [22]. Apparently, despite being doped and embedded in a polymeric matrix, hydroxyquinoline does not lose its transparency at high wavelengths.

The optical band gap is the most relevant parameter to evaluate the potential of (Alq_3_-TCNQ):Ppy film for optoelectronic devices as active layers. In this work, the optical band gap (*E_g_^opt^*) was calculated through Tauc’s semi-empirical model [68]. Such a method is based on the relation [68,69] $\alpha h\nu =B{\left(h\nu -E_{g}^{opt}{}_{\;}\right)}^{n}$, where h is Planck’s constant, B is a parameter that depends on the probability of transition, $E_{g}^{opt}{}_{\;}$ is the optical band gap and n is a number that characterizes the transition process, where *n* = 2 for indirect allowed transitions. The absorption coefficient (α) and frequency (ν) are experimentally obtained from $\alpha =ln\left(T/d\right)$ and $\nu =\frac{c}{\lambda }$, respectively. In these expressions, *T* is the transmittance, *d* is the thickness of the film, *c* is the speed of light and *λ* is the wavelength. The thickness of 8500 Å was measured by profilometry. The dependence of ${\left(\alpha h\nu \right)}^{n}$ on hν was plotted and *E_g_^opt^* was evaluated from the *x*-axis intercept at (αhν)^1/2^ = 0. The values of $E_{g}^{opt}{}_{\;}$ calculated for the films are shown in Figure 6c. The (Alq_3_-TCNQ):Ppy shows two transitions; the first one at 2.58 eV is the onset gap (*E_g_^onset^*) and the second one at 3.06 eV corresponds to the optical gap (*E_g_^opt^*) [70]. This first transition corresponds to a bound electron–hole pair’s optical absorption and formation (Frenkel exciton) [69,70,71]. The second transition is the energy gap between the valence band (π-band) and the conduction band (π*-band) [71]. The electronic transitions from π to π* explain *E_g_^onset^* and *E_g_^opt^* as a consequence of several factors, including defects, film morphology, structural disorder and traps. However, the obtained value of 2.58 eV is less than the band gaps from 2.66 to 2.84 eV for pristine Alq_3_ [22,24,26]; it is also less than 3.97 eV for ZnO-doped Alq_3_ [18] and 3.2 eV for Alq_3_ doped with sexithiophene [72]. The presence of TCNQ and Ppy decreases the optical gap and also generates the onset gap, which is an indication of a more efficient semiconductor behavior in the (Alq_3_-TCNQ):Ppy film with respect to other semiconductors derived from Alq_3_, such as the ones mentioned above.

On the other hand, the Urbach energy (*E_U_*) can be used to determine the defects in the band gap and can be determined according to the equation [73,74] $\alpha =A_{a}exp\left(\frac{hv}{E_{U}}\right)$, where, in addition to the parameters defined above, *A_a_* is a constant of the material that corresponds to the α at the energy gap. The exponential absorption edge can be interpreted as due to the exponential distribution of local states in the energy band gap [73]. Figure 6d displays the linear relation between ln(α) and hν for the hybrid films. The value of the *E_U_* was determined from the reciprocal of the slope from this linear relation, and, in this case, it is 0.717 eV. Considering that *E_U_* is zero in a perfect semiconductor, the obtained value is high due to the bulk heterojunction between hydroxyquinoline, its dopant and the polymer. However, the interconnections between these three components of the hybrid film can generate charge transport both at the edges of Alq_3_ and the TCNQ and at the interface of the (Alq_3_-TCNQ):Ppy film with the electrodes, where charges are less common and efficient when evaluating their electrical behavior. The number of satisfactorily dissociated charges depends on the magnitude of the internal field generated and is limited by the charge mobility of each material.

After studying the band gap, a photoluminescence (PL) analysis of the sample was carried out at room temperature (see Figure 7). It can be seen that, after exciting with a 325 nm laser (UV), the sample shows an intense visible emission in the visible region, with a maximum at around 550 nm. Other groups have observed similar tendencies for the fluorescence or PL of 8-Hydroxyquinoline and their derivatives [75]. A higher intensity of these samples has been attributed to longer alkyl chains, and the mechanism can be mainly attributed to excited state proton transfer (ESPT).

### 4.4. Fabrication and Electrical Characterization of the Device

In order to evaluate the electrical behavior in the (Alq_3_-TCNQ):Ppy film, Figure 8 shows the I-V curves for the FTO/(Alq_3_-TCNQ):Ppy/Ag device under the different types of used light. In Figure 8a, it can be seen that the behavior of the current is similar for all types of used light. In a visual fashion, an ohmic regime is appreciated from −1.2 V to −1 V, while the interval −1 V to 0 V shows another ohmic regime. An SCLC regime can be found from 0 V to 0.6 V, and, after 0.6 V, another ohmic regime is appreciated. Considering a cross-section area of $5.77{\;\mathrm{mm}}^{2},$ the current density–voltage (J-V) curves of the film are shown in Figure 8b; the same regimes can be seen in Figure 8a. As observed in Figure 8a,b, both the current and the current density are less in the dark condition, which means that it exhibits a photoconductive effect, and its conductivity increased due to the applied light. Since the continuous functions that describe the current densities of the device are unknown, it is not possible to determine the exact values of voltage where the regime transitions occur. The cubic spline method [76] has been applied to each J-V curve to approximate a tendency to the data from Figure 8b to fit cubic polynomials to the current density data. From cubic polynomials, it is easy to determine inflection points where changes in the concavity of the function take place and, therefore, in the current density. At some of these points, the ohmic to SCLC transition voltage $V_{\mathrm{ON}}$ and the trap-filled-limit voltage $V_{\mathrm{TFL}}$ are determined.

The cubic polynomials are defined by SJV=JiV=a3iV3+a2iV2+a1iV+a0i with i=0,…,k−1. Notice that the spline function is defined as SJV:a,b→ℝ, where a=V0≤V1≤…≤Vk−1≤Vk=b and V=V0,V1,…,Vk is the data vector; furthermore, each polynomial JiV is defined as JiV:Vi,Vi+1→ℝ. Moreover, a Vi∗ point is an inflection point if SJ′Vi∗=0 and SJ″Vi∗>0. Applying the cubic spline approach and the conditions for determining inflection points, 491 inflection points were found; the histogram of the voltage values where an inflection point is localized is shown in Figure 9.

From Figure 9, the quartiles were set at 13, 56 and 70 percent of the data because the most significant concavity changes are presented in those datasets. Then, the voltage values where the most significant inflection points of the J-V curve were found are −0.88 V, 0.16 V and 0.54 V. Therefore, the voltage intervals that define the different behaviors of the J-V curve are $I_{1}\in \left[-1.2,-0.9\right]$, $I_{2}\in \left[-0.9,\;0.16\right]$ and $I_{3}\in \left[0.16,\;0.56\right].$ Based on these intervals, $V_{\mathrm{ON}}=0.16\;$V and $V_{\mathrm{TFL}}=0.56\;$V. The approximation curves of the ohmic and SCLC regimes as well as the transition voltages are shown in Figure 10. From Figure 10, the red curve represents the ohmic regime, which was estimated by a straight line; a cyan curve is the quadratic SCLC regime, and it can be seen that $V_{\mathrm{ON}}$ joins both these regimes. On the other hand, $V_{\mathrm{TFL}}$ delimits the end of the quadratic SCLC regime. Since after the $V_{\mathrm{TFL}}$ an ohmic regime was observed (magenta curve), it was also approximated by a straight line.

Figure 10 also shows the J-V curve of the device under dark conditions (blue curve). The behavior of the J-V curves on the intervals can be observed. $I_{1}$, $I_{2}$ and $I_{3}$ are similar when exposed to dark conditions and to different types of light. On the other hand, it can be seen that the slopes of the second ohmic zone, after the $V_{\mathrm{TFL}}$ voltage, are different. When the applied voltage is greater than $V_{\mathrm{TFL}}=0.56$ V in the dark condition, the current density flow is lower than that of the light condition. Conversely, the current density flow increases when the applied voltage is greater than $V_{\mathrm{TFL}}=0.56\;$V and the device is exposed to UV-Vis light. This means that the device conducts better when exposed to light. With the approximation of the ohmic to SCLC transition voltage $V_{\mathrm{ON}}$ and the trap-filled-limit voltage $V_{\mathrm{TFL}}$ and Equations (3) and (4), the free carrier density $n_{0}$ and the trap density $N_{t}$ can be determined as follows.
(5)$$n_{0}=\frac{9V_{\mathrm{ON}}\varepsilon_{0}\varepsilon_{r}}{8qd_{s}^{2}}$$
(6)$$N_{t}=\frac{2V_{\mathrm{TFL}}\varepsilon_{r}}{qd_{s}^{2}}$$

Since $V_{\mathrm{ON}}=0.16$ V, $V_{\mathrm{TFL}}=0.56$ V, $q=1.6\times 10^{-19}C$, $\varepsilon_{0}=8.85418\times 10^{-12}\frac{C}{\mathrm{Vm}}$ $d_{s}=8.5\times 10^{-7}m$ and assuming a dimensionless dielectric constant $\varepsilon_{r}=3.234,$ an approximation of the free carrier density and trap density for the device are $n_{0}=4.4586\times 10^{19}\frac{1}{m^{3}}$ and $N_{t}=3.1333\times 10^{31}\frac{1}{m^{3}}$, respectively.

## 5. Conclusions

In recent years, optimizing the organic semiconductor manufacturing process has been a topic of scientific interest. In these optimization processes, new organic semiconductors are used, and then it is necessary to have methodologies that allow to quickly obtain the electrical characteristics of these materials to determine if the device will have the best performance. In this sense, the hybrid film (Alq_3_-TCNQ):Ppy was fabricated by initially doping the hydroxyquinoline with the TCNQ and subsequently embedding the semiconductor in the polypyrrole polymer matrix. DFT calculations carried out on designed complexes of these species show that the nature of these complexes is determined by the formation of hydrogen bonds, which involve two possible cases: (i) the interaction among nitrogen terminal atoms of TCNQ moiety with peripheral hydrogen atoms from the Ppy and (ii) face to face interactions between TCNQ central hydrogen atoms and the heteroatoms from the organic rings of Ppy. The formation of these complexes rules the decrease in the HOMO-LUMO gap, as shown by the calculations. The experimental IR spectra indicate that the hybrid film did not suffer degradation during its deposit. The SEM and AFM images have confirmed that the film shows a heterogeneous morphology integrated by segregation of the Alq_3_-TCNQ semiconductor in the matrix, and the X-ray diffraction results confirmed that the clusters assembled in the film and that the film itself exhibits an amorphous structure. The UV-Vis spectrum of the (Alq_3_-TCNQ):Ppy film showed a characteristic band at 375 nm, attributed to the electron transition from the HOMO orbital in hydroxyquinoline to the LUMO orbital in the TCNQ. The optical properties of the (Alq_3_-TCNQ):Ppy film showed two indirect allowed transitions, with onset and optical gaps of 3.44 and 2.64 eV. Moreover, the photoluminescence measurements showed an intense visible emission in the visible region. By applying the cubic spline method to the J-V curves of the device, it was determined that the ohmic to SCLC transition voltage is $V_{\mathrm{ON}}=0.16$ V and the trap-filled-limit voltage is $V_{\mathrm{TFL}}=0.56$ V; then, considering that the dielectric constant is $\varepsilon_{r}=3.234$, the free carrier density and trap density of the device were calculated, yielding $n_{0}=4.4586\times 10^{19}\frac{1}{m^{3}}$ and $N_{t}=3.1333\times 10^{31}\frac{1}{m^{3}}$, respectively. Future work includes the determination of the charge carrier mobility of the device. From the J-V curves of the device, it was possible to verify that the current density is higher when the device is exposed to UV light, meaning the resistance to current flow decreased and, consequently, its conductivity increased. On the other hand, under dark conditions, the current density was lower, which means the device exhibits greater resistance to the passage of current, and therefore its conductivity decreased. The shape of the J-V curves demonstrates that the device has a photoconductive effect. These results indicate that the (Alq_3_-TCNQ):Ppy film can be used as a photoconductor in optoelectronic applications.

## Figures and Tables

**Figure 1 sensors-23-07708-f001:**
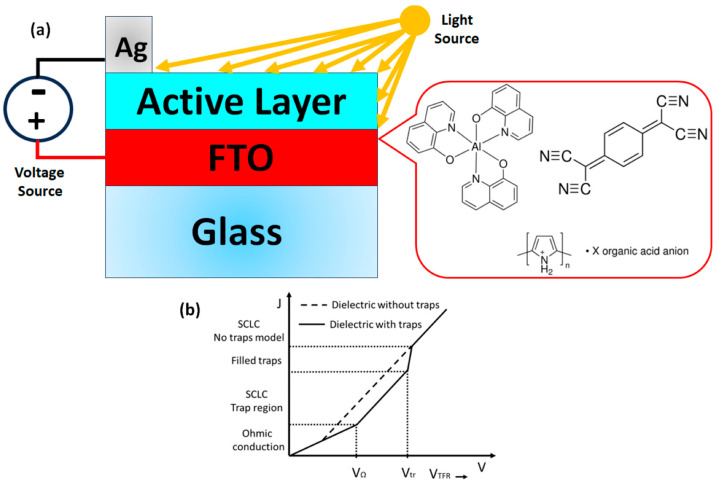
(**a**) Diagram of the photoconductor device for electrical characterization. (**b**) Current density-voltage characteristic [59].

**Figure 2 sensors-23-07708-f002:**
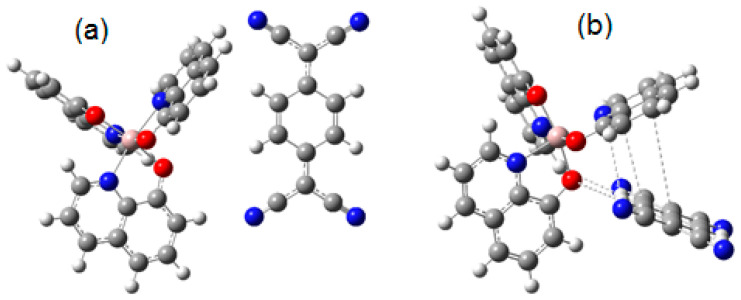
Optimized structures of (**a**) Hal-TCNQ1 and (**b**) Hal-TCNQ2. (Code color: carbon, grey; oxygen, red; nitrogen, blue and pink aluminium).

**Figure 3 sensors-23-07708-f003:**
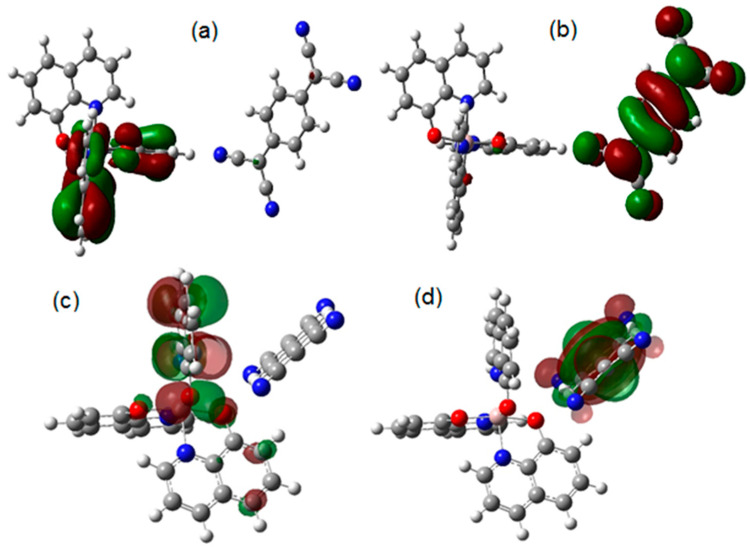
Frontier molecular orbitals for hAl-TCNQ1: (**a**) HOMO, (**b**) LUMO and hAl-TCNQ2, (**c**) HOMO and (**d**) LUMO. (The green and red colors represent the positive and negative phases of the molecular orbitals respectively).

**Figure 4 sensors-23-07708-f004:**
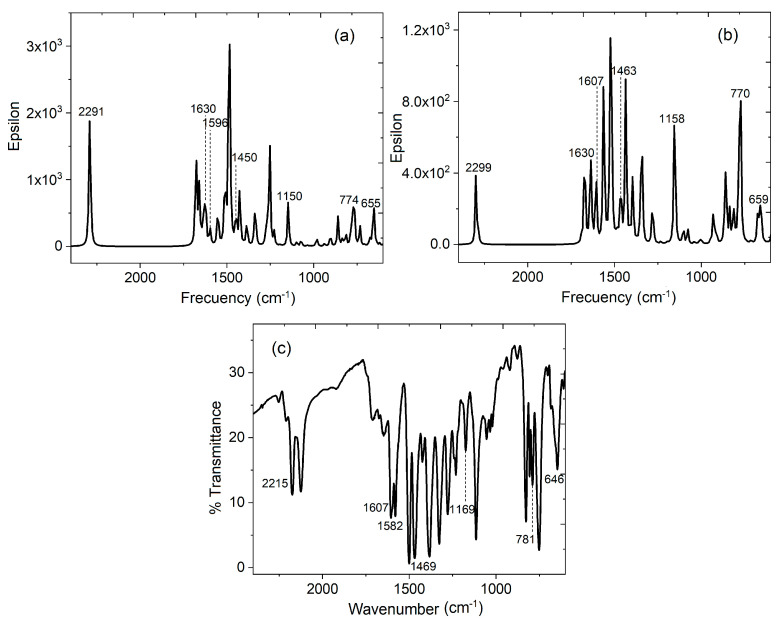
Calculated IR spectrum for (**a**) hAl-TCNQ1, (**b**) hAl-TCNQ2 and (**c**) experimental IR spectrum for Alq_3_-TCNQ in KBr pellet.

**Figure 5 sensors-23-07708-f005:**
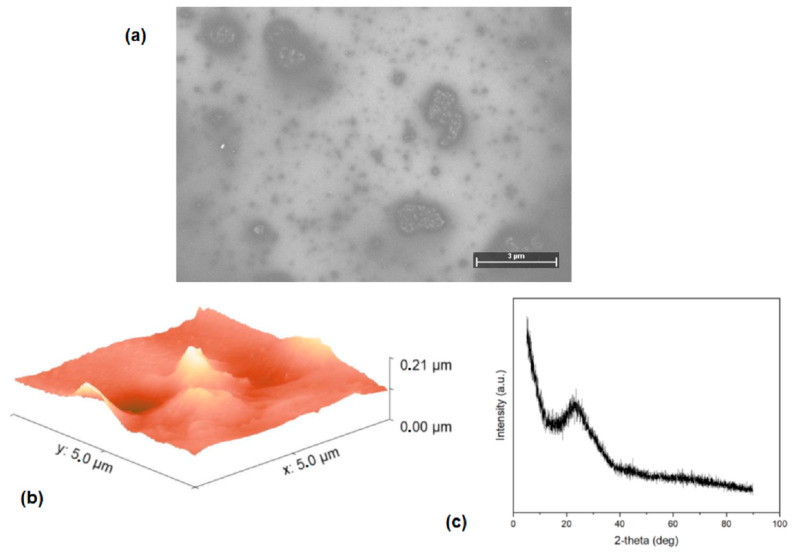
(**a**) SEM, (**b**) DRX pattern and (**c**) AFM measurements of the (Alq_3_-TCNQ):Ppy film.

**Figure 6 sensors-23-07708-f006:**
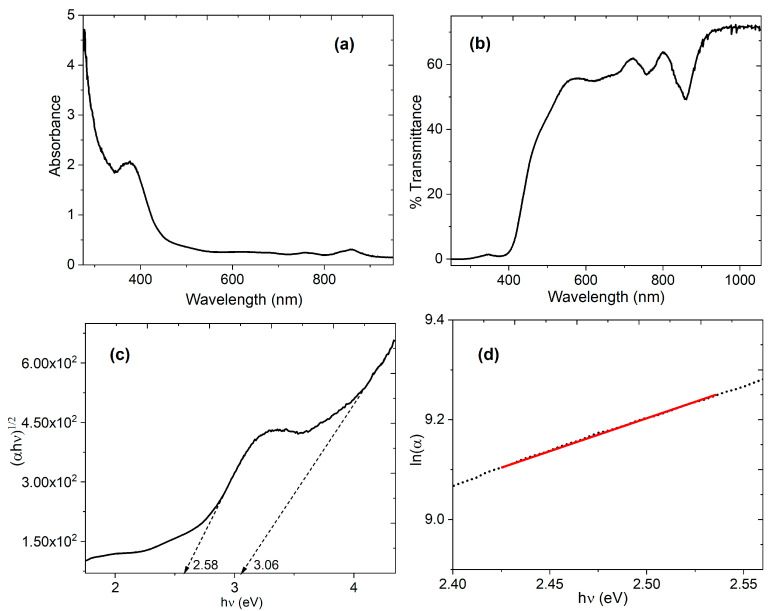
(**a**) Absorbance spectra, (**b**) % transmittance, (**c**) Tauc plot and (**d**) variation in ln(α) with hν for (Alq_3_-TCNQ):Ppy film. The slope for the Urbach energy was calculated on the red line.

**Figure 7 sensors-23-07708-f007:**
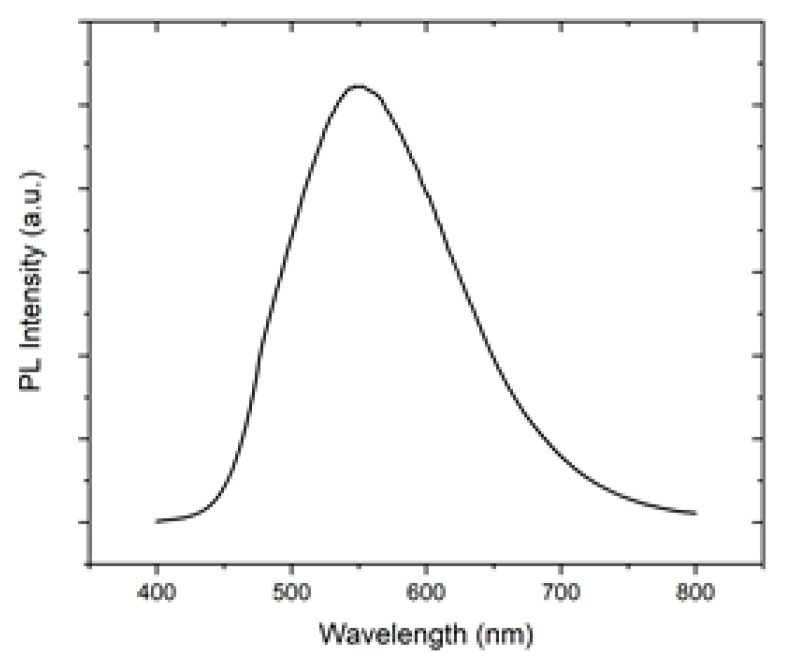
Photoluminescence spectra of the (Alq_3_-TCNQ):Ppy film at room temperature with a UV laser.

**Figure 8 sensors-23-07708-f008:**
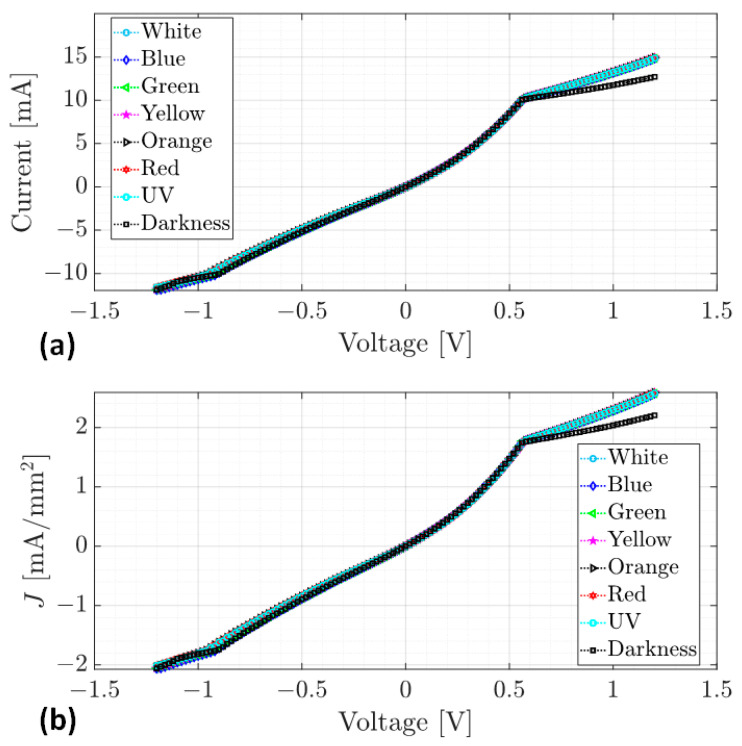
(**a**) Current–voltage curves and (**b**) current density–voltage curves of FTO/(Alq_3_-TCNQ):Ppy/Ag device.

**Figure 9 sensors-23-07708-f009:**
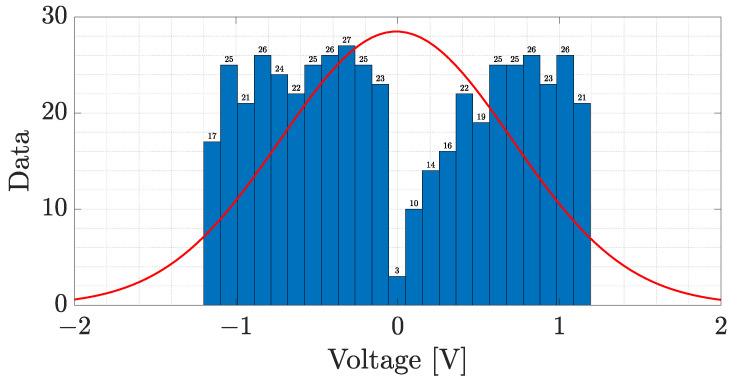
Inflection points histogram.

**Figure 10 sensors-23-07708-f010:**
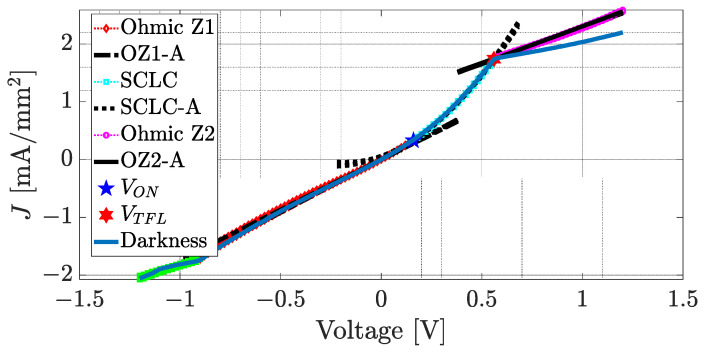
Approximation of the J-V graphs.

**Table 1 sensors-23-07708-t001:** Distance and Wiberg index of the interacting atoms in hAl-TCNQ1 and hAl-TCNQ2.

Distance (Å)				Wiberg Index
Alq_3_	TCNQ			hAl-TCNQ1
	N (62)	N (53)		
H (22)		2.85		0.0018
H (23)		2.76		0.0024
H (49)	2.49			0.0074
Alq_3_	TCNQ			hAl-TCNQ2
	N (53)	H (69)	H (70)	
H (15)	2.69			0.0036
O (32)		3.04		0.0023
O (51)			2.079	0.0257

**Table 2 sensors-23-07708-t002:** Calculated and experimental IR frequencies and their assignment for Alq_3_-TCNQ.

hAl-TCNQ1 *ν* (cm^−1^) Simulated	hAl-TCNQ2 *ν* (cm^−1^) Simulated	Alq_3_-TCNQ *ν* (cm^−1^) KBr Pellet	(Alq_3_-TCNQ):Ppy *ν* (cm^−1^) Film	Assignment
1630	1630	1607	1609	C=C
1596	1607	1582	1591	C=N
1450	1463	1469	1466	C-C
1150	1158	1169	1156	C-C-H
774, 655	770, 659	781, 646	772, 648	in-plane ring deformations
2291	2299	2215	2217	CN-stretching mode

## Data Availability

Data are contained within the article.

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
