# Peer review of "Preparation of Hybrid Films Based in Aluminum 8-Hydroxyquinoline as Organic Semiconductor for Photoconductor Applications"

_sensors, 2023, doi:10.3390/s23187708_

Round 1

Reviewer 1 Report

The authors systematically investigated the optical and electrical properties of (Alq3-TCNQ):PPy films by theoretical and experimental characterizations. The doping of organic semiconductors are always an interesting topic in organic electronics. However, the following issues should be addressed before being consideration for publication in the journal Sensors:

1. The authors claimed that their hybrid films are for organic light-emitting devices in the title and introduction. However, I cannot see any related evidences including experiments (e.g., EL spectra) to support this in the manuscript. 

2. Line 92-93, what does it mean? i.e, "there is a high contact surface between Alq3 and TCNQ, which favors the diffusion of excitons." Contact surface usually favors the discossiation of excitons. How does it favor the diffusion?

3. More experimental details should be provided. How to deposit the films? dip coating? How thick are their films (electrode thickness, active layers, etc.)?

4. What is the motivation to invesigate the current under light illumination with different wavelengths? Which side does the light come from? From Ag side? How about light intensity and dark current in Fig. 7?

5. There are many typos in the present manuscript, such as "whit" in Line 126, "it" in Line 186, "MFA" in 269...

Author Response

Comments and Suggestions for Authors

The authors systematically investigated the optical and electrical properties of (Alq3-TCNQ):PPy films by theoretical and experimental characterizations. The doping of organic semiconductors are always an interesting topic in organic electronics. However, the following issues should be addressed before being consideration for publication in the journal Sensors:

  1. The authors claimed that their hybrid films are for organic light-emitting devices in the title and introduction. However, I cannot see any related evidences including experiments (e.g., EL spectra) to support this in the manuscript. 

ANSWER. Thanks for your observation. The study of photoluminescence was included, and the manuscript was modified towards the type of device consistent with the results obtained in this work: a phototransistor.

  1. Line 92-93, what does it mean? i.e, "there is a high contact surface between Alq3 and TCNQ, which favors the diffusion of excitons." Contact surface usually favors the discossiation of excitons. How does it favor the diffusion?

ANSWER. Thanks for the remark, this statement has been corrected. In the layer with the bulk heterojunction architecture, a high contact surface is generated between the polypyrrole (PPy) matrix and the organic semiconductor Alq3-TCNQ. This favors the dissociation of excitons.

  1. More experimental details should be provided. How to deposit the films? dip coating? How thick are their films (electrode thickness, active layers, etc.)?

ANSWER. The information regarding the methodology used to deposit the films was completed. On the other hand, the thickness was measured by profilometry, details about this measurement have been included in the experimental methodology.

  1. What is the motivation to invesigate the current under light illumination with different wavelengths? Which side does the light come from? From Ag side? How about light intensity and dark current in Fig. 7?

ANSWER. The motivation to study the current under different lighting conditions was included in the manuscript. This study is related to determining if the optical properties of the hybrid film affect its electrical behavior.

On the other hand; Figure 1 was modified in order to visualize which side the light is coming from. The light source is located at the top of the device.

Finally, in the section on Fabrication and Electrical Characterization of the device, the I-V graph and the discussion of the results on the electrical behavior of the device, under dark current conditions, were included.

  1. There are many typos in the present manuscript, such as "whit" in Line 126, "it" in Line 186, "MFA" in 269...

ANSWER. The manuscript was carefully checked and typographical errors corrected.

Reviewer 2 Report

The authors synthesized tris(8-hydroxyquinoline) aluminum doped with tetracyanoquinodimethane to be used as a photoactive layer in organic light-emitting device. This work has some merit, however it requires improvements.

1/ There are some errors to correct.

2/ The authors should add XRD diffractogram of the active layer to access its structural properties and then compare it with DFT results.

3/ For example, to calculate the absorption coefficient, the authors need to use the thickness of the film. How to control this thickness during the film growth? The authors should add the SEM cross section to verify the obtained thickness.

4/ It is recommended to add photoluminescence spectrum for this active layer especially will be used for OLED.

Moderate editing of English language are required.

Author Response

Comments and Suggestions for Authors

The authors synthesized tris(8-hydroxyquinoline) aluminum doped with tetracyanoquinodimethane to be used as a photoactive layer in organic light-emitting device. This work has some merit, however it requires improvements.

1/ There are some errors to correct.

ANSWER. The manuscript was carefully reviewed and errors corrected.

2/ The authors should add XRD diffractogram of the active layer to access its structural properties and then compare it with DFT results.

ANSWER. The XRD diffractogram of the active layer was added. However; the structure is amorphous and the comparison of it with theoretical calculations is not possible, because the calculation were carried out on single molecules and it would be comparable with single crystal X-ray spectroscopy but it is not the present case.

3/ For example, to calculate the absorption coefficient, the authors need to use the thickness of the film. How to control this thickness during the film growth? The authors should add the SEM cross section to verify the obtained thickness.

ANSWER. Thanks for your observation. We have measured the thickness by profilometry. Information on thickness measurement has been included in the experimental methodology section of the manuscript.

4/ It is recommended to add photoluminescence spectrum for this active layer especially will be used for OLED.

ANSWER. The photoluminescence spectrum was included.

Comments on the Quality of English Language

Moderate editing of English language are required.

ANSWER. The manuscript was carefully checked and corrected for spelling and grammar in the English language.

Round 2

Reviewer 1 Report

Though the authors have provided new experimental data, I am still not convinced since my concerns have not been addressed. 

1. Electroluminescence was not provided.

2. "Organic light-emitting Devices" in the title has been modified as "Organic Phototransistors", however, the authors did not provide experimental data for organic phototransistors. The device they studied is a sandwich-type photodiode, instead of a phototransistor. 

3. In Lines 38 and 39, I cannot agree with the definition of "photoconductor" consisting of photoresistors, photodiodes and phototransistors. The author should cite any classic literatures or books to support the definition.

4. For phototransistor or photodiodes, the authors should perform corresponding characterizations for photoresponsivity, detectivity, mobility and so on.

5. There are still some typos in the manuscript, e.g., DRX in Line 300.

Author Response

REVIEWER 1

Comments and Suggestions for Authors

Though the authors have provided new experimental data, I am still not convinced since my concerns have not been addressed. 

  1. Electroluminescence was not provided.

RESPONSE. Thank you for your query. In the present manuscript, we want to design noble nanostructures with interesting morphological, electrical, and optical properties. As reported, we have already checked the photoluminescence, and the intense-visible emission embarks the potential application of the present material. However, EL, as you mentioned, is quite important, but it is out of the scope of the current work, and we have planned future manuscripts for this remaining work.

  1. "Organic light-emitting Devices" in the title has been modified as "Organic Phototransistors", however, the authors did not provide experimental data for organic phototransistors. The device they studied is a sandwich-type photodiode, instead of a phototransistor. 

RESPONSE. Thanks for check this out.

We have reviewed and discussed some works such as

  • Yoo, H., Lee, I. S., Jung, S., Rho, S. M., Kang, B. H., Kim, H. J., A Review of Phototransistors Using Metal Oxide Semiconductors: Research Progress and Future Directions. Adv. Mater. 2021, 33, 2006091. https://doi.org/10.1002/adma.202006091
  • Chow, P.C.Y., Matsuhisa, N., Zalar, P. et al. Dual-gate organic phototransistor with high-gain and linear photoresponse. Nat Commun 9, 4546 (2018). https://doi.org/10.1038/s41467-018-06907-6
  • Ross, D.A. (1979). Solid State Photodetectors — The Photoconductor. In: Optoelectronic Devices and Optical Imaging Techniques. Palgrave, London. https://doi.org/10.1007/978-1-349-16219-2_3
  • Ross, D.A. (1979). Solid State Photodetectors — The Photodiode and Phototransistor. In: Optoelectronic Devices and Optical Imaging Techniques. Palgrave, London. https://doi.org/10.1007/978-1-349-16219-2_4

So, based on the works and the experimental current-voltage and current density-voltage curves (see figures 8(a), 8(b) and 10), the device is a photoconductor because the incident light causes an increase in the current flow, compared to darkness conditions. Furthermore, since the photoconductor device is a sandwich-type and has only two connection terminals, we agree that the photoconductor device is more like a photodiode instead a phototransistor.

Based on the experimental curve of figure 10, it is important to highlight the following characteristics of the photoconductor device:

  • The current density flow of the device is the same in both light and darkness conditions when the applied voltage is between -1.2 and 0.56 volts.
  • When the applied voltage is greater than 0.56 volts and there is darkness condition, the current density flow is lower than light condition.
  • The current density flow increases when the applied voltage is greater than 0.56 volts and the device is exposed to UVis light.

Strictly speaking, under the different types of light used, the J-V curves of the device are not identical to the typical curves of a photodiode. So, we have decided to use the term photoconductor.

  1. In Lines 38 and 39, I cannot agree with the definition of "photoconductor" consisting of photoresistors, photodiodes and phototransistors. The author should cite any classic literatures or books to support the definition.

RESPONSE. Thank you for this suggestion. We agree with this modification.

At the last version, the state of the art on organic optocouplers and photodetectors was taken from the following articles:

  1. Sun, Q., Dong, G., Wang, L., Qiu, Y. Organic optocouplers. Sci. China Chem. 2011, 54, 1017–1026. https://doi.org/10.1007/s11426-011-4283-1
  2. Baeg, K.-J., Binda, M., Natali, D., Caironi, M. and Noh, Y.-Y. Organic Light Detectors: Photodiodes and Phototransistors. Adv. Mater., 2013, 25, 4267-4295. https://doi.org/10.1002/adma.201204979
  3. Seeds, A.J. & De Salles, A.A.A., Optical control of microwave semiconductor devices, IEEE Transactions on Microwave Theory and Techniques, 1990, 38, 577-585. doi: 10.1109/22.54926.

This new version added the following reference:

  1. Ross, D.A. Solid State Photodetectors — The Photoconductor. In: Optoelectronic Devices and Optical Imaging Techniques. Palgrave, London, 1979. https://doi.org/10.1007/978-1-349-16219-2_3

Reference 7 was taken from the book:

Ross, D.A. Optoelectronic Devices and Optical Imaging Techniques, Red Globe Press London, 1979.

The new version of the paragraph is next:

Speaking specifically of organic optocoupler (OOC), it has three main parts: organic photoemitter, insulator and organic photodetector [4]. In a particular way, the organic photodetector or photosensor can be divided into organic photoresistors, organic photodiodes or organic phototransistors [4,5]. These organic semiconductor devices have a photoconductive effect, which modifies their conductivity due to the incidence of light, i.e., if a voltage is applied, a current will flow and the incident light on the device will cause an increase in this current [6,7]. The current flow in a semiconductor in the absence of some kind of light is called dark current, whereas the current flow produced from an incident light is called photon induced current or photocurrent [7]. When an organic semiconductor has the photoconductive effect may be referred to organic photoconductor (OPC).

  1. For phototransistor or photodiodes, the authors should perform corresponding characterizations for photoresponsivity, detectivity, mobility and so on.

RESPONSE. Thank you for this suggestion.

We agree that with the characterization of the photoresponsivity, detectivity and mobility of the device, the quality of the work would be higher. However, these procedures are considered as future work on applications.

  1. There are still some typos in the manuscript, e.g., DRX in Line 300.

RESPONSE. The manuscript was carefully checked and typographical errors corrected.

Reviewer 2 Report

The authors corrected the manuscript and improved it as requested. This work can be considered for publication in Sensors.

Minor editing of English language are required.

Author Response

Minor editing of English language are required.

RESPONSE. The manuscript was carefully checked and typographical errors corrected.

Round 3

Reviewer 1 Report

Since my conserns have been addressed, I can recommend it for publication after minor revisions:

1. Line 25, please delete "type".

2. Make the lines in Fig. 8 more distinguishable. 

Author Response

REVIEWER No. 1

Comments and Suggestions for Authors

Since my conserns have been addressed, I can recommend it for publication after minor revisions:

  1. Line 25, please delete "type".

Answer. The word "type" was removed

  1. Make the lines in Fig. 8 more distinguishable. 

Answer. Thanks for the observation. Figure 8 was modified and the lines are more distinguishable.
